# Understanding Higher-Order Interactions in Information Space

**DOI:** 10.3390/e26080637

**Published:** 2024-07-27

**Authors:** Herbert Edelsbrunner, Katharina Ölsböck, Hubert Wagner

**Affiliations:** 1ISTA (Institute of Science and Technology Austria), 3400 Klosterneuburg, Austria; edels@ist.ac.at (H.E.); katharina.oelsboeck@ist.ac.at (K.Ö.); 2Department of Mathematics, University of Florida, Gainesville, FL 32611, USA

**Keywords:** higher-order interactions, topological data analysis, persistent homology, simplicial complex, alpha shape, wrap complex, information theory, Shannon entropy, relative entropy, Bregman divergence, non-Euclidean geometry, Bregman geometry

## Abstract

Methods used in topological data analysis naturally capture higher-order interactions in point cloud data embedded in a metric space. This methodology was recently extended to data living in an information space, by which we mean a space measured with an information theoretical distance. One such setting is a finite collection of discrete probability distributions embedded in the probability simplex measured with the relative entropy (Kullback–Leibler divergence). More generally, one can work with a Bregman divergence parameterized by a different notion of entropy. While theoretical algorithms exist for this setup, there is a paucity of implementations for exploring and comparing geometric-topological properties of various information spaces. The interest of this work is therefore twofold. First, we propose the first robust algorithms and software for geometric and topological data analysis in information space. Perhaps surprisingly, despite working with Bregman divergences, our design reuses robust libraries for the Euclidean case. Second, using the new software, we take the first steps towards understanding the geometric-topological structure of these spaces. In particular, we compare them with the more familiar spaces equipped with the Euclidean and Fisher metrics.

## 1. Introduction

The motivation for the work reported in this paper is the deeper understanding of the effect of the ambient metric on the higher-order interactions arising in data, as well as its multi-scale geometric–topological properties. The first can be viewed via the Čech or Alpha complex, the second through their persistent homology. Specifically, we generalize geometric and topological data analysis methods from Euclidean geometry to Bregman geometries in which dissimilarity is measured with divergences that are generally non-symmetric and violate the triangle inequality. By necessity, these methods are sensitive to the dissimilarity defining the ambient geometry, and we exploit this sensitivity to quantify the difference between geometries.

As example geometries, we emphasize those related to information-theoretic concepts, such as the Shannon geometry and the Fisher geometry. What we call the Shannon geometry is induced by the relative entropy (Kullback–Leibler divergence) [1], which is based on Shannon’s entropy [2]. The Fisher geometry is induced by the Fisher distance (more technically the Fisher–Rao metric [3]). It was introduced by Rao [4], building on Fisher’s seminal work [5] in statistics.

These are examples of what we call information spaces  [6]. Such spaces often exhibit unfamiliar behavior, different from the Euclidean space we are used to. Understanding such spaces is however important, as—plainly speaking—this is where many types of data live [7].

Importantly, these geometries are commonly used in data science. For example, relative entropy is a standard loss function used in dimensionality reduction [8,9] and deep learning  [7,10]. (This connection may not be immediately obvious. Many models aim to explicitly minimize what is called cross-entropy. It has the same minimizer as the relative entropy, which can additionally be viewed as a distance, and therefore gives rise to a geometry). The Fisher geometry starts to be considered [11] as an alternative.

We therefore deem it important to shed light on the differences and commonalities between these geometries. Some pertinent questions are as follows: Is the simpler Fisher geometry a good approximation of the Shannon geometry? Can we see a significant difference between the Euclidean geometry and the non-Euclidean ones, as predicted by the discrepancy in their practical performances?

In the current work, we are particularly interested in the algorithms that underpin the data analysis methods, especially the topological ones. While the Fisher geometry can be handled with Euclidean tools [6], the Shannon geometry used to require customized ones [12]. We show that the Shannon geometry can also benefit from existing robust tools, although in this case the application is less direct. We also hope that this development opens new alleys for topological data analysis in information spaces.

**Prior work and results.** The research reported in this paper merges several lines of work. One is the study of Bregman divergences from the computational geometry point of view initiated in [13] and the extension of topological data analysis to Bregman and Fisher geometries started in [6,12]. Another is the study of higher-order interactions in high-dimensional probabilistic feature spaces [14,15], which generally lack information-theoretical interpretations.

Pivotal work at the intersection of computational geometry and Bregman geometry includes the extension of k-means clustering [16], proximity search data structures [17,18], and Voronoi tessellations [13] to the Bregman setting.

On a practical note, we mention a research direction focusing on detecting, loosely speaking, the holes in knowledge [14,15,19,20] understood as a collection of text documents. In the simplest case, each text document is represented as a discrete probability distribution (by counting characteristic keywords [21]). Interestingly, higher-order interactions among documents are necessary to capture the multi-scale topology of these data. Indeed, pairwise, triplewise, etc., intersections between balls (with respect to the appropriate metric or divergence) centered at the data points reveal the topological structure.

In this work, we will also use computational topology methods. One important topological tool is the nerve and the related Nerve Theorem. Intuitively, it allows one to capture the topology (more precisely, the homotopy type) of a union of (a finite collection of) objects using a finite simplicial complex. The theorem is usually attributed to Borsuk [22] or Leray [23], and is stated more explicitly by Weil [24]. See [25] for a comprehensive, modern treatment of this topic.

A particularly important concept in our investigations is the Bregman–Delaunay mosaic, which we formally define as the straight-line dual of the not necessarily straight-line Bregman–Voronoi tessellation obtained by measuring distance with the Bregman divergence from a data point. This mosaic was already defined in [13], and we explain how it can be computed as a weighted Euclidean Delaunay mosaic using standard geometric software. In Euclidean space, the Alpha shapes can be defined as the sublevel sets of the radius function on the Delaunay mosaic, which is a generalized discrete Morse function in the sense of Forman [26] and Freij [27]. The lower sets of the critical simplices of this function constitute the Wrap complex [28], which was introduced as a shape reconstruction tool in [29]. We extend this framework by introducing the rise function on a Bregman–Delaunay mosaic, which provides a convenient measure of the size of a Bregman ball. With these notions, we construct the shape of data in different geometries, and we use them to quantify the difference between the geometries.

We have implemented all the algorithms and used the resulting software to run experiments comparing the Euclidean, Shannon, and Fisher geometries for synthetic data. We find that the Delaunay mosaics and their Alpha and Wrap complexes in these geometries show some, occasionally subtle, differences, which we quantify.

**Outline.** Section 2 provides the necessary background from discrete geometry and computational topology. Section 3 gives the details needed to compute Delaunay mosaics and their Alpha and Wrap complexes in Bregman and Fisher geometries using software for weighted Delaunay mosaics in Euclidean geometry. Section 4 presents computational experiments, and Section 5 discusses the quantification of the difference between Bregman and other geometries. Section 6 concludes the paper. In Appendix A, Appendix B, Appendix C and Appendix D, we provide supplementary experiments, results, and background information.

## 2. Background

We need some background on Bregman divergences, Delaunay mosaics, and discrete Morse functions. Indeed, this paper combines these concepts to obtain new insights into Bregman–Delaunay mosaics and their scale-dependent subcomplexes.

**Bregman divergence.** Given a suitable convex function on a convex domain, the best affine approximation at a point defines a dissimilarity measure on the domain (see [30]). We follow [31] in the details of this construction, requiring a technical third condition that guarantees a conjugate function of the same kind. Let $\Omega \subseteq R^{d}$ be an open and convex domain. A function F:Ω→R is of Legendre type [32] if

(i)*F* is differentiable;(ii)*F* is strictly convex;(iii)∇F diverges whenever we approach the boundary of Ω.

If the boundary of the domain is empty, which is the case for $\Omega =R^{d}$, then Condition (iii) is void. In other words, ∥∇F(x)∥ does not necessarily diverge when ∥x∥→∞. Given points x,y∈Ω, the Bregman divergence from *x* to *y* associated with *F* is the difference between *F* and the best affine approximation of *F* at *y*, both evaluated at *x*:(1)$$\begin{array}{cc} D_{F}\left(x∥y\right) & =F\left(x\right)-\left[F(y)+\langle \nabla F(y),x-y\rangle \right], \end{array}$$
where 〈.,.〉 denotes the standard dot product. Note that $D_{F}\left(x∥y\right)\geq 0$, with equality iff x=y, resembling a metric. However, the other two axioms of a metric do not hold: the divergence is not necessarily symmetric, and it violates the triangle inequality in all non-trivial cases. In spite of these shortcomings, Bregman divergences are useful as measures of dissimilarity and are sufficient to define a geometry.

For a given h≥0, the primal ball with center *x* contains all points *y* such that the divergence from *x* to *y* is at most *h*, and the dual ball contains all points *y* such that the divergence from *y* to *x* is at most *h*: (2)$$\begin{array}{cc} B_{F}\left(x,h\right) & =\{y\in \Omega |D_{F}\left(x∥y\right)\leq h\}, \end{array}$$(3)$$\begin{array}{cc} B_{F}^{*}\left(x,h\right) & =\{y\in \Omega |D_{F}\left(y∥x\right)\leq h\}. \end{array}$$
The geometric intuition for (Equation 2) is to cast light onto the graph of *F* from a point vertically above $x\in R^{d}$ in $R^{d+1}$ and at distance *h* below the graph of *F*: the primal ball is the vertical projection of the lit up part of the graph onto $R^{d}$. This ball is not necessarily convex. The geometric intuition for (3) is to intersect the graph of *F* with the tangent hyperplane at *x* shifted vertically upward by a distance *h*: the dual ball is the vertical projection of the part of the graph on or below this shifted hyperplane. This ball is necessarily convex.

The conjugate of *F* can be constructed with elementary geometric means. Specifically, we use the polarity transform that maps a point $A=\left(a,a_{d+1}\right)\in R^{d}\times R$ to the affine map $A^{*}:R^{d}\to R$ defined by $A^{*}\left(x\right)=\langle a,x\rangle -a_{d+1}$. Similarly, it maps $A^{*}$ to $A={\left(A^{*}\right)}^{*}$. The graph of *F* can be described as a set of points or a set of affine maps that touch the graph. The conjugate function, $F^{*}:\Omega^{*}\to R$, is defined such that polarity maps the points of the graph of *F* to the tangent affine maps of the graph of $F^{*}$, and it maps the tangent affine maps of the graph of *F* to the points of the graph of $F^{*}$. Since *A* and $A^{*}$ switch position with gradient, so do *F* and $F^{*}$. More specifically, $\Omega^{*}=\phi \left(\Omega \right)$ and $F^{*}:\Omega^{*}\to R$ are given by
(4)$$\begin{array}{cc} \phi (x) & =\nabla F(x), \end{array}$$
(5)$$\begin{array}{cc} F^{*}\left(\phi \left(x\right)\right) & =\langle \nabla F(x),x\rangle -F(x), \end{array}$$
(6)$$\begin{array}{cc} \nabla F^{*}\left(\phi \left(x\right)\right) & =x. \end{array}$$
The convexity of Ω and Conditions (i), (ii), (iii) imply that $\Omega^{*}$ is convex and $F^{*}$ satisfies (i), (ii), (iii). In other words, the conjugate of a Legendre type function is again a Legendre type function. Importantly, the Bregman divergences associated with *F* and with $F^{*}$ are symmetric: $D_{F}\left(x∥y\right)=D_{F^{*}}\left(\phi (y)∥\phi (x)\right)$. Hence, ϕ maps primal balls to dual balls and it maps dual balls to primal balls:(7)$$\begin{array}{cc} B_{F^{*}}^{*}\left(\phi (x),h\right) & =\phi (B_{F}\left(x,h\right)), \end{array}$$(8)$$\begin{array}{cc} B_{F^{*}}\left(\phi (x),h\right) & =\phi (B_{F}^{*}\left(x,h\right)). \end{array}$$
Since all dual balls are convex, all primal balls are diffeomorphic images of convex sets. This implies that the common intersection of a collection of primal balls is either empty or contractible [12], so the Nerve Theorem applies.

**Examples.** An important example of a Legendre type function is $\varpi :R^{d}\to R$ defined by mapping *x* to half the square of its Euclidean norm: $\varpi \left(x\right)=\frac{1}{2}{∥x∥}^{2}$. It is the only Legendre type function that is its own conjugate: $\varpi =\varpi^{*}$. The symmetry between the divergences of a Legendre type function and its conjugate thus implies $D_{\varpi }\left(x∥y\right)=D_{\varpi }\left(y∥x\right)$ and $B_{\varpi }\left(x,h\right)=B_{\varpi }^{*}\left(x,h\right)$. Indeed, it is easy to see that the divergence is half the squared Euclidean distance, $D_{\varpi }\left(x∥y\right)=\frac{1}{2}{∥x-y∥}^{2}$, which is of course symmetric. This particular Legendre type function provides an anchor point for comparison.

The example that justifies the title of this paper is the (negative) Shannon entropy [2]. We add an extra term, resulting in $E:R_{+}^{d}\to R$, defined as $E\left(x\right)=\sum_{i=1}^{d}\left[x_{i}\ln x_{i}-x_{i}\right]$. Being linear, the additional term does not affect the resulting divergence, but simplifies certain computations later.

This function is of Legendre type and fundamental to information theory [2]. Its divergence,
(9)$$\begin{array}{cc} D_{E}\left(x∥y\right) & =\sum_{i=1}^{d}\left[x_{i}\ln x_{i}-x_{i}\ln y_{i}-x_{i}+y_{i}\right], \end{array}$$
is generally referred to as the relative entropy [10] or the Kullback–Leibler divergence [1] from *x* to *y*. We remark that the above derivation using the Bregman machinery results in a divergence which is valid on the entire positive orthant of $R^{d}$, and agrees with the standard definition [1] on the standard simplex.

The gradient of the Shannon entropy at *x* is the vector ∇E(x) with components $\ln x_{i}$ for 1≤i≤d. Using Equation (5), one easily computes that the conjugate of *E* maps this vector to $\sum_{i=1}^{d}x_{i}$. Hence $E^{*}:R^{d}\to R$ is defined by mapping $y\in R^{d}$ to $E^{*}\left(y\right)=\sum_{i=1}^{d}e^{y_{i}}$.

A case of special interest is the restriction of the Shannon entropy to the standard simplex, which is a subset of the positive orthant. Writing $x=(x_{1},x_{2},\ldots ,x_{d})$ for a point of $R_{+}^{d}$, the *standard (d−1)-simplex*, denoted $\Delta^{d-1}$, consists of all points *x* that satisfy $x_{1}+x_{2}+\ldots +x_{d}=1$. We use $\Delta^{d-1}$ as the domain of a Legendre type function, which is the reason why we introduce $\Delta^{d-1}$ as an open set. Finally, write $E_{\Delta }:\Delta^{d-1}\to R$ for the restriction of the Shannon entropy to the standard simplex. This setting is important because each $x\in \Delta^{d-1}$ can be interpreted as a probability distribution on *d* disjoint events. Correspondingly, $-E_{\Delta }\left(x\right)=-E\left(x\right)$ is the expected efficiency to optimally encode a sample from this distribution. Finally, the relative entropy from *x* to *y* is the expected loss in coding efficiency if we use the code optimized for *y* to encode a sample from *x*. Projecting the gradient of the unrestricted Shannon entropy into the hyperplane of the simplex passing through the origin, we obtain the gradient of the restriction:(10)$$\begin{array}{cc} \nabla E_{\Delta }\left(x\right) & =\left[\begin{array}{c} \ln x_{1}\\ \ln x_{2}\\ \vdots \\ \ln x_{d} \end{array}\right]-\frac{1}{d}\sum_{i=1}^{d}\ln x_{i}\left[\begin{array}{c} 1\\ 1\\ \vdots \\ 1 \end{array}\right]. \end{array}$$
Using (Equation 4) and (5), we compute the conjugate of $E_{\Delta }$, which we state in terms of the barycentric coordinates parametrizing $R^{d-1}$. Specifically, we obtain $\phi_{\Delta }\left(x\right)=\nabla E_{\Delta }\left(x\right)$ and
(11)$$\begin{array}{cc} E_{\Delta }^{*}\left(\phi_{\Delta }\left(x\right)\right) & =\langle \nabla E_{\Delta }\left(x\right),x\rangle -E_{\Delta }\left(x\right) \end{array}$$
(12)$$\begin{array}{cc} \; & =1-\frac{1}{d}\sum_{i=1}^{d}\ln x_{i} \end{array}$$
(13)$$\begin{array}{cc} \; & =1+\ln\sum_{i=1}^{d}e^{y_{i}}, \end{array}$$
in which the $y_{i}=\ln x_{i}-\frac{1}{d}\sum_{i=1}^{d}\ln x_{i}$ are the coordinates in conjugate space. Indeed, it is not difficult to verify (13) using $\ln\sum_{i=1}^{d}x_{i}=0$ for points in the standard simplex.

**Antonelli isometry.** A Bregman divergence gives rise to a path metric in which length is measured by integrating the square root of the divergence. As explained in [6], any divergence that decomposes into a term per coordinate implies an isometry between this path metric and the Euclidean metric. By (Equation 9), the relative entropy is an example of such a divergence, and the corresponding path metric is known as the Fisher metric, which plays an important role in statistics and information geometry [3]. Instead of formalizing the recipe for constructing the Fisher metric from the relative entropy, we note that the mapping
(14)$$\begin{array}{cc} ȷ(x) & =(\sqrt{2x_{1}},\sqrt{2x_{2}},\ldots ,\sqrt{2x_{d}}) \end{array}$$
is an isometry with Euclidean space, as first observed by Antonelli [33]. By virtue of being an isometry, the distance between points $x,y\in R_{+}^{d}$ under the Fisher metric satisfies ${∥x-y∥}_{\mathrm{Fsh}}=∥ȷ(x)-ȷ(y)∥$. The path of this length from *x* to *y* is the preimage of the line segment from ȷ(x) to ȷ(y), which is generally not straight.

Of special interest is the Fisher metric restricted to the standard (d−1)-simplex, denoted $\Delta^{d-1}$. The mentioned isometry maps $\Delta^{d-1}$ to $ȷ(\Delta^{d-1})$, which is the positive orthant of the sphere with radius $\sqrt{2}$ and center at the origin in $R^{d}$. The shortest path between $x,y\in \Delta^{d-1}$ is thus the preimage of the great-circle arc that connects ȷ(x) and ȷ(y) on the sphere. Since this arc is generally longer than the straight line segment connecting ȷ(x) and ȷ(y) in $R_{+}^{d}$, the distance between *x* and *y* under the Fisher metric restricted to $\Delta^{d-1}$ is generally larger than in the unrestricted case.

**Alpha shapes and Wrap complexes.** Two popular shape reconstruction methods based on Delaunay mosaics are the Alpha shapes introduced in [34] and the Wrap complexes first published in [29]. Both extend to generalized discrete Morse functions and therefore to Bregman–Delaunay mosaics and Bregman–Wrap complexes. While working with Bregman divergences, we can construct them using weighted Euclidean Delaunay mosaics, for which there is readily available fast software. For brevity, standard definitions and properties are available in Appendix B. Letting *D* be a simplicial complex and f:D→R a generalized discrete Morse function, the Alpha complex for *h* is the sublevel set,
(15)$$\begin{array}{cc} {\mathrm{Alpha}}_{h}\left(f\right) & =f^{-1}\left(-\infty ,h\right], \end{array}$$
and the Alpha shape is the underlying space of the Alpha complex. In contrast to the Alpha shape, the assumption that *f* be a generalized discrete Morse function is essential in the definition of the Wrap complex. Recall that every step of a generalized discrete Morse function is an interval of simplices in the Hasse diagram. We form the step graph, $G=G_{f}$, whose nodes are the steps and whose arcs connect step φ to step ψ if there are simplices P∈φ and Q∈ψ with an arc from *P* to *Q* in the Hasse diagram. By construction, *f* is strictly increasing along directed paths in the step graph, which implies that the graph is acyclic.

The lower set of a node ν in G, denoted ↓ν, is the set of nodes φ for which there are directed paths from φ to ν. Similarly, we write $↓\;N={⋃}_{\nu \in N}↓\;\nu$ for the lower set of a collection of nodes, and ⋃↓N for the corresponding collection of simplices. We are particularly interested in the set of singular intervals, and we recall that each such interval contains a critical simplex of *f*. We write ${\mathrm{Sg}}_{f}$ for the set of singular intervals, and ${\mathrm{Sg}}_{f}\left(h\right)\subseteq {\mathrm{Sg}}_{f}$ for the subset whose simplices satisfy f(Q)≤h. The Wrap complex for *h* is the union of steps in the lower sets of the singular intervals with value at most *h*:(16)$$\begin{array}{cc} {\mathrm{Wrap}}_{h}\left(f\right) & =⋃↓\;{\mathrm{Sg}}_{f}\left(h\right). \end{array}$$
There are alternative constructions of the Wrap complex. Starting with the Alpha complex for *h*, we obtain the Wrap complex for the same value by collapsing all non-singular intervals that can be collapsed. The order of the collapses is not important as all orders produce the same result, namely ${\mathrm{Wrap}}_{h}\left(f\right)$. Symmetrically, we may start with the critical simplices of value at most *h* and add the minimal collection of non-singular intervals needed to obtain a simplicial complex. The minimal collection is unique and so is the result, ${\mathrm{Wrap}}_{h}\left(f\right)$. A proof of the equivalence of these three definitions of the Wrap complex is given in Appendix C.

## 3. Mosaics and Algorithms

In this section, we review Bregman–Delaunay and Fisher–Delaunay mosaics as well as their scale-dependent subcomplexes. All mosaics are constructed using software for weighted Delaunay mosaics in Euclidean geometry; all subcomplexes are computed by convex optimization, adapting the method from [12]. We begin with the mosaics in Bregman geometry.

**Bregman–Delaunay mosaics.** Let $\Omega \subseteq R^{d}$ be open and convex, consider a Legendre type function F:Ω→R, and let U⊆Ω be locally finite. Following [12,13], we define the Bregman–Voronoi domain of u∈U, denoted ${\mathrm{dom}}_{F}\left(u,\Omega \right)$, as the points a∈Ω that satisfies $D_{F}\left(u∥a\right)\leq D_{F}\left(v∥a\right)$ for all v∈U. The Bregman–Voronoi tessellation is the collection of such domains, and the Bregman–Delaunay mosaic records all non-empty common intersections:(17)$$\begin{array}{cc} {\mathrm{Vor}}_{F}\left(U,\Omega \right) & =\{{\mathrm{dom}}_{F}\left(u,\Omega \right)|u\in U\}, \end{array}$$(18)$$\begin{array}{cc} {\mathrm{Del}}_{F}\left(U,\Omega \right) & =\{Q\subseteq U|{⋂}_{u\in Q}{\mathrm{dom}}_{F}\left(u,\Omega \right)\neq \emptyset \}, \end{array}$$
and we note that the mosaic is isomorphic to the nerve [35,36] of the tessellation. To develop geometric intuition, we observe that ${\mathrm{Vor}}_{F}\left(U,\Omega \right)$ can be obtained by growing primal Bregman balls with centers at the points u∈U. When two such balls meet, they freeze where they touch but keep growing everywhere else. Eventually, each ball covers exactly the corresponding domain. Since the primal balls are not necessarily convex, it is not surprising that the faces shared by the domains are not necessarily straight. Nevertheless, the Delaunay mosaic has a natural straight-line embedding as all its cells are vertical projections of the lower faces of the convex hull of the points $\left(u,F\left(u\right)\right)\in R^{d+1}$. To see this, we note that each cell of the mosaic corresponds to a dual Bregman ball whose boundary passes through the vertices of the cell, and this ball is the vertical projection of the part of the graph of *F* on or below the graph of an affine function.

**Construction.** To construct the mosaic, we assume that U⊆Ω is in general position, by which we mean that Conditions I and II from Appendix D are satisfied after transforming U⊆Ω to $X\subseteq R^{d}\times R$ such that ${\mathrm{Del}}_{F}\left(U,\Omega \right)$ is a subcomplex of the weighted Delaunay mosaic of *X*. Lifting the points from $R^{d}$ to $R^{d+1}$ and projecting the lower boundary of the convex hull back to $R^{d}$, we obtain the mosaic. We remind the reader that relevant background information can be found in Appendix B, and define $\varpi \left(a\right)=\frac{1}{2}{∥a∥}^{2}$. We formalize this method while stating all steps in terms of weighted points in *d* dimensions:Step 1: Let $X\subseteq R^{d}\times R$ be the set of weighted points x(u)=(u,2ϖ(u)−2F(u)), with u∈U.Step 2: Compute the weighted Delaunay mosaic of *X* in Euclidean geometry, denoted Del(X).Step 3: Select ${\mathrm{Del}}_{F}\left(U,\Omega \right)$ as the collection of simplices in Del(X) whose corresponding weighted Voronoi cells have a non-empty intersection with $\Omega^{*}$.

Indeed, the weighted Delaunay mosaic computed in Step 2 may contain simplices that do not belong to the Delaunay–Bregman mosaic of *F*. To implement Step 3, we note that ${\mathrm{Del}}_{F}\left(U,\Omega \right)$ is dual to ${\mathrm{Vor}}_{F}\left(U,\Omega \right)$, which is isomorphic to ${\mathrm{Vor}}_{F^{*}}\left(\phi \left(U\right),\Omega^{*}\right)$, and this Bregman–Voronoi tessellation is the weighted Voronoi tessellation of *X* restricted to $\Omega^{*}$. This tessellation has convex polyhedral cells and is readily available as the dual of Del(X). Writing Y(Q)⊆X for the points x(u) with u∈Q⊆U and dom(Y) for the weighted Voronoi cell that corresponds to Y∈Del(X), we have
(19)$$\begin{array}{cc} {\mathrm{Del}}_{F}\left(U,\Omega \right) & =\{Q\subseteq U|\mathrm{dom}\left(Y(Q)\right)\cap \Omega^{*}\neq \emptyset \}. \end{array}$$
Instead of computing all these intersections, we can collapse Del(X) to the desired subcomplex and thus save time by looking only at a subset of the mosaic. We explain how the simplices can be organized to facilitate such a collapse. Recalling that $\Omega^{*}\subseteq R^{d}$ is open and convex, we introduce the signed distance function, $\theta :R^{d}\to R$, which maps every $a\in R^{d}$ to plus or minus r=r(a) such that the sphere with center *a* and radius *r* touches $\partial \Omega^{*}$ but does not cross the boundary. Finally, θ(a)=r(a) if $a\notin \Omega^{*}$ and θ(a)=−r(a) if $a\in \Omega^{*}$. Note that $\Omega^{*}=\theta^{-1}\left[-\infty ,0\right)$ and that $\Omega_{t}^{*}=\theta^{-1}\left[-\infty ,t\right)$ is open and convex for every *t*. Now construct ϑ:Del(X)→R by mapping Y∈Del(X) to the maximum t∈R for which $\mathrm{dom}\left(Y\right)\cap \Omega_{t}^{*}=\emptyset$. By (Equation 19), we obtain ${\mathrm{Del}}_{F}\left(U,\Omega \right)$ by removing all simplices *Y* with ϑ(Y)≥0. The crucial observation is that for *X* in general position, ϑ is a generalized discrete Morse function with a single critical vertex. To see this, we observe that Vor(X) decomposes $\Omega_{t}^{*}$ into convex domains for every value *t*, which by the Nerve Theorem [22] implies that $\vartheta^{-1}\left(-\infty ,t\right]$ is contractible. Removing the simplices in a sequence of decreasing values of ϑ thus translates into a sequence of collapses that preserve the homotopy type of the mosaic.

**Rise functions.** To introduce scale into the construction of Bregman–Delaunay mosaics, we generalize the radius function from Euclidean geometry to Bregman geometries, changing the name because size is more conveniently measured by height difference in the (d+1)-st coordinate direction as opposed to the radius in $R^{d}$. Let $\dot{u}=\left(u,F\left(u\right)\right)$ and $\bar{u}:R^{d}\to R$ be the point and affine map that correspond to u∈Ω, and let $\upsilon :R^{d}\to R$ be the upper envelope of the $\bar{u}$, u∈U. We introduce the rise function, $\varrho_{F}:{\mathrm{Del}}_{F}\left(U,\Omega \right)\to R$, which maps each simplex, *Q*, to the minimum difference between $F^{*}$ and υ at points in the conjugate Voronoi cell:(20)$$\begin{array}{cc} \varrho_{F}\left(Q\right) & =\underset{a\in \phi (\mathrm{dom}\left(Q,\Omega \right))}{\inf}\left[F^{*}\left(a\right)-\upsilon \left(a\right)\right]. \end{array}$$
It is the infimum amount we have to lower the graph of $F^{*}$ until it intersects the graph of υ at a point vertically above the Voronoi cell in conjugate space. Without going to the conjugate, we can interpret $\varrho_{F}\left(Q\right)$ in terms of (primal) Voronoi domains and cones of light cast from the $\dot{u}$ onto the graph, which we raise until the cones clipped to within their Voronoi domains have a point in common. This interpretation motivates the name of the function. Comparing (Equation 20) with (Equation 32), we see that the two agree when F=ϖ and $\Omega =R^{d}$. Indeed, we obtain $F^{*}=\varpi$ and υ=ξ. Furthermore, ϕ(dom(Q,Ω))=dom(Q,Ω), and taking the infimum is the same as taking the minimum.

For every h∈R, we have a sublevel set, ${\mathrm{Del}}_{F,h}\left(U,\Omega \right)={\varrho_{F}}^{-1}\left(-\infty ,h\right]$, which we refer to as the Bregman–Alpha complex of *U* and *F* for size *h*. For h<0, this complex is empty, for h=0, it is a set of vertices namely the points in *U*, and for sufficiently large positive *h*, this complex is ${\mathrm{Del}}_{F}\left(U,\Omega \right)$.

**Information-theoretic interpretations.** We offer a brief information-theoretical interpretation for many of the geometric objects mentioned above in the case of relative entropy.

Let us start with a primal relative entropy ball of radius *r* around a finite probability distribution *c*. It contains all distributions that can be used to approximate *c*, incurring the efficiency loss of at most *r* bits. With this, it is easy to see that the Voronoi and Delaunay constructions have intuitive information-theoretical interpretations. For simplicity, let us consider the Čech rise (radius) function, which arises from the first point of a non-empty intersection between a number of primal balls [35,36]. This radius is therefore the smallest number of bits that need to be incurred when approximating the centers of these balls with a single distribution. This value can be interpreted as an information-theoretical quantification of the high-order interactions among these distributions. Of course, the intersection point is exactly the unique best choice for such a distribution.

The remaining objects can be interpreted in a similar way, which we leave for the reader. We stress that the birth and death values in the resulting persistence diagrams also express the loss of coding efficiency in bits—and stress that the diagrams are identical for the Delaunay and Čech case.

**Computation.** We compute the rise function following the intuition based on primal Voronoi domains explained below (Equation 20). Equivalently, $\varrho_{F}\left(Q\right)$ is the minimum amount we have to raise the graph of *F* so it has a supporting hyperplane that passes through all points $\dot{u}$, with u∈Q, while all other point $\dot{u}$, with u∈U, lie on or above the hyperplane.

To turn this intuition into an algorithm, we consider the affine hull of *Q* and write $\bar{v}:\mathrm{aff}\;Q\to R$ for the affine function that satisfies $\bar{v}\left(u\right)=F\left(u\right)$ for all u∈Q. Let H:affQ∩Ω→R measure the difference: $H\left(a\right)=F\left(a\right)-\bar{v}\left(a\right)$. Since *F* is of Legendre type, so is *H*. We are interested in the infimum of *H*, which either occurs at a point in affQ∩Ω or at the limit of a divergent sequence. We therefore introduce a numerical routine that returns both, the infimum and the point where it occurs: 1InfSize (functionF,simplexQ):2      $\left(a_{Q},h_{Q}\right)=\left(\mathrm{arginf}\;H,\inf H\right)$;3      return $\left(a_{Q},h_{Q}\right)$.


Note that the dual Bregman ball centered at aQ∈affQ∩Ω of size hQ contains *Q* in its boundary, and it may or may not contain points of U\Q in its interior. If it does not, then ϱF(Q)=hQ, otherwise, ϱF(Q) is the minimum function value of the proper cofaces of *Q*. To express this more formally, we write coFacets(Q) for the collection of simplices R∈Del(X) with Q⊆R and #R=#Q+1. Since *Q* gets its value either directly or from a coface, it is opportune to compute the rise function in the order of decreasing dimension:1for p=d downto 0 do2      forall *p*-simplices Q∈DelF(U,Ω) do3            (aQ,hQ) = InfSize(*F*, *Q*);4            if BF*(aQ,hQ)∩[U\Q]=∅5                  then ϱF(Q)=hQ6                  else ϱF(Q)=minR∈coFacets(Q)ϱF(R).

Note that this algorithm assigns a value to every simplex in ${\mathrm{Del}}_{F}\left(U,\Omega \right)$. Indeed, the simplices in Del(X) that are not in ${\mathrm{Del}}_{F}\left(U,\Omega \right)$ have been culled in Step 3, as explained above.

**Fisher metric.** In addition to the Bregman divergences, we consider Delaunay mosaics under the Fisher metric. To construct them, we recall that the mapping $ȷ:R_{+}^{d}\to R_{+}^{d}$ defined by $ȷ\left(x\right)=\left(\sqrt{2x_{1}},\sqrt{2x_{2}},\ldots ,\sqrt{2x_{d}}\right)$ is an isometry between the Fisher metric and the Euclidean metric. This suggests the following algorithm:Step 1: Compute the Delaunay mosaic of ȷ(U) in Euclidean space.Step 2: Remove the simplices from Del(ȷ(U)) whose dual Voronoi cells have an empty intersection with $R_{+}^{d}$.Step 3: Draw the resulting complex by mapping each point ȷ(u) to the original point $u\in U\subseteq R_{+}^{d}$.

The rise function in Euclidean geometry maps every simplex ȷ(Q)∈Del(ȷ(U)) to the squared radius of the smallest empty circumsphere of ȷ(Q). By isometry, the preimage of this Euclidean sphere is the smallest empty circumsphere of *Q* under the Fisher metric, and the squared radius is the same. We thus obtain the rise function on the Fisher–Delaunay mosaic by copying the values of the rise function on the Delaunay mosaic in Euclidean geometry.

The construction of the mosaic for the Fisher metric restricted to the standard simplex, $\Delta^{d-1}$, is only slightly more complicated. As mentioned in Section 2, the isometry maps $\Delta^{d-1}$ to $\sqrt{2}S_{+}^{d-1}$, which is our notation for the positive orthant of the sphere with radius $\sqrt{2}$ centered at the origin in $R^{d}$. The distance between points $u,v\in \Delta^{d-1}$ under the Fisher metric thus equals the Euclidean length of the great-circle arc connecting $ȷ\left(u\right),ȷ\left(v\right)\in \sqrt{2}S_{+}^{d-1}$. The Delaunay mosaic of ȷ(U) under the geodesic distance can be obtained by constructing the convex hull of ȷ(U)∪{0} in $R^{d}$ and centrally projecting all faces not incident to 0 onto the sphere. As before, we cull simplices whose dual Voronoi cells have an empty intersection with the positive orthant of the sphere, and we draw the mosaic in $\Delta^{d-1}$ by mapping the vertices back to the original points. Furthermore, the rise functions of the mosaics in $\sqrt{2}S_{+}^{d-1}$ and in $\Delta^{d-1}$ are the same. Note, however, that the geodesic radius is the arc-sine of and therefore slightly larger than the straight Euclidean radius in $R^{d}$.

## 4. Computational Experiments

We illustrate the Bregman–Alpha and Bregman–Wrap complexes while comparing them to the conjugate, the Fisher, and the Euclidean constructions.

**Example in positive quadrant.** Let *X* be a set of 1000 points uniformly distributed according to the Fisher metric in ${\left(0,2\right]}^{2}\subseteq R_{+}^{2}$. To sample *X*, we use the isometry, $ȷ:R_{+}^{2}\to R_{+}^{2}$, between the Fisher and the Euclidean metric mentioned in Section 2. Specifically, we sample 1000 points uniformly at random according to the Euclidean metric in ${\left(0,2\right]}^{2}$, and we map each point with coordinates $x_{1}$, $x_{2}$ to ${ȷ}^{-1}\left(x_{1},x_{2}\right)=\frac{1}{2}\left(x_{1}^{2},x_{2}^{2}\right)$, which is again a point in ${\left(0,2\right]}^{2}$. To compute the Delaunay mosaic in Fisher geometry, we construct the (Euclidean) Delaunay mosaic of ȷ(X) and draw this mosaic with the vertices at the points in *X*. Recall however that the domain is $\Omega =R_{+}^{d}$ and not $R^{d}$. A simplex whose corresponding Voronoi cell has an empty intersection with the positive orthant thus does not belong to the mosaic, which is restricted to Ω. We identify these simplices and remove them from the Delaunay mosaic as described in Section 3.

**Example in standard triangle.** Motivated by our interest in information-theoretic applications, we repeat the above experiment within the standard triangle, $\Delta^{2}$, which consists of all points $\left(x_{1},x_{2},x_{3}\right)\in R_{+}^{3}$ that satisfy $x_{1}+x_{2}+x_{3}=1$. Every point in $\Delta^{2}$ can be interpreted as a probability distribution on three disjoint events, which is indeed the most relevant scenario for the application of the relative entropy. To sample a set *Y* of 1000 points uniformly at random according to the Fisher metric in $\Delta^{2}$, we use again j, now restricted to $\Delta^{2}$, whose image is the positive orthant of the sphere with radius $\sqrt{2}$ centered at the origin of $R^{3}$. Sampling 1000 points uniformly at random according to the geodesic distance on the sphere, we take the convex hull of ȷ(Y)∪{0} and obtain the mosaic by mapping the vertices to the points in $Y={ȷ}^{-1}\left(ȷ\left(Y\right)\right)$. Before drawing the faces in $\Delta^{2}$, we remove 0 and all incident faces, as well as all faces whose corresponding Voronoi cells have an empty intersection with $R_{+}^{2}$.

Recall that the squared Fisher metric matches the relative entropy in the infinitesimal regime, which explains the random appearance of the reconstruction in Figure 1, for which we set the threshold to 0.0025. The reconstruction in Shannon geometry is similar to those in conjugate Shannon geometry in Figure 1a and in Fisher geometry in Figure 1b. To interpret the reconstruction in Figure 1d, we observe that the difference between the Shannon entropy and the squared Euclidean norm has a minimum at the center and no other critical points in the interior of the triangular domain. Accordingly, the reconstruction removes simplices near the corners and the three sides first. More drastically, the Bregman–Wrap complex for the same data removes all simplices except for a single critical edge near the center (see Figure 2d).

## 5. Quantification of Difference

We take a data-centric approach to quantifying the differences between the geometries. Given a common domain, Ω, and a finite set of points, U⊆Ω, we compare the corresponding mosaics and rise functions.

**Mosaics.** The Delaunay mosaics of *U* depend on the local shape of the balls defined by the metric or the divergence. Letting *D* and *E* be two Delaunay mosaics with vertex sets *U*, we compare them by counting the common cells:(21)$$\begin{array}{cc} J(D,E) & =1-\frac{\#\left(D\cap E\right)}{\#D+\#E-\#\left(D\cap E\right)}, \end{array}$$
which is sometimes referred to as the *Jaccard distance* between the two sets. It is normalized so that J=0 iff D=E and J=1 iff *D* and *E* share no cells at all. In our application, the two mosaics share all vertices, so *J* is necessarily strictly smaller than 1. To apply this measure to the constructions in Section 4, we write $D_{0},\;D_{1},\;D_{2},\;D_{3},\;D_{4}$ for the mosaics in Figure A1, and we write $E_{0},\;E_{1},\;E_{2},\;E_{3},\;E_{4}$ for the mosaics in Figure 3. All mosaics are different, except for $D_{0}=D_{4}$ and $E_{0}=E_{4}$. The Jaccard distances are given in Table 1.

We see that the mosaics in conjugate Shannon geometry and in Fisher geometry are most similar to each other and less similar to the mosaic in Shannon geometry. The mosaic in Euclidean geometry is most dissimilar to the others (see Figure 3 and related Figure A1 in Appendix A for visual confirmation).

**Rise functions.** Different rise functions on the same mosaic can be compared by counting the inversions, which are the pairs of cells whose orderings are different under the two functions. Recall that $D_{0}=D_{4}$ and $E_{0}=E_{4}$, let $d_{0}:D_{0}\to R$ and $e_{0}:\;E_{0}\to R$ be the rise functions in Shannon geometry, and let $d_{4}:\;D_{4}\to R$ and $e_{4}:\;E_{4}\to R$ be the rise functions in weighted Euclidean geometry. The normalized number of inversions are
(22)$$\begin{array}{cc} I(d_{0},d_{4}) & =0.476, \end{array}$$
(23)$$\begin{array}{cc} I(e_{0},e_{4}) & =0.467. \end{array}$$
In words, slightly fewer than half the pairs are inversions, both for $d_{0},d_{4}$ and for $e_{0},e_{4}$. This is plausible because $d_{4}$ orders the cells along the diagonal while $d_{0}$ preserves the random character of the point sample (see Figure A2d). Similarly, $e_{4}$ orders the cells radially, from the center of the standard triangle to its periphery, while $e_{0}$ preserves again the random character of the sample (see Figure 4d).

We can compare the rise functions also visually, by color-coding the 2-dimensional cells, and this works even if the mosaics are different. Specifically, we shade the triangles by mapping small to large rise function values onto dark to light colors. In Figure A2a,b, this leads to randomly mixed dark and light triangles, while in Figure A2c,d, there are clear but opposing gradients parallel to the diagonal. Similarly, in Figure 4c, we see the rise function decreasing from the center to the boundary of the standard triangle, and in Figure 4d we see it increasing from the center to the boundary. In addition, we compare general rise functions by computing their persistence diagrams (see [36]). Writing Dgm(d) for the persistence diagram of function *d*, we quantify the difference with the bottleneck between the diagrams:(24)$$\begin{array}{cc} B(d,e) & =W_{\infty }\left(\mathrm{Dgm}\left(d\right),\mathrm{Dgm}\left(e\right)\right). \end{array}$$
As explained in [36], the bottleneck distance is 1-Lipschitz, that is: $B\left(d,e\right)\leq {∥d-e∥}_{\infty }$, but d≠e does not necessarily imply B(d,e)≠0. The bottleneck distances between the $d_{i}:\;D_{i}\to R$ and the $e_{i}:\;E_{i}\to R$ are given in Table 2.

In part, this comparison agrees with the Jaccard distances between the mosaics given in Table 1. The most obvious disagreements are for $d_{0},d_{4}$ and for $e_{0},e_{4}$, in which quite different functions are defined on identical mosaics.

## 6. Discussion

We extended two popular computational geometric-topological methods within the framework of discrete Morse functions and showed how this generalizes the methods to data in Bregman and Fisher geometries. Importantly, this allowed us to produce a robust implementation without the need to develop customized low-level software. We hope that this result will provide extra motivation for the development of geometric software for high-dimensional Euclidean space, as such software can be reused to handle data in information space. In particular, efficient computation of skeleta of weighted Delaunay mosaics would be relevant for topological analysis of high-dimensional data.

Turning the table, we use these generalized methods to compare different geometries experimentally. Our experimental approach to studying geometries is a first step towards an intuitive understanding of these often counter-intuitive geometries, as well as the high-order interactions occurring in them.

We reiterate one reason why understanding such geometries is important: they underpin many modern data science methods. It is, for example, a prerequisite for explaining the surprisingly good performance of many deep learning models (that output predictions as discrete probability distributions). More broadly, the space of discrete probability distributions is a basic object in mathematics, but is not well understood. We provide new tools for broadening the understanding of such spaces.

Perhaps the main observation coming from our experiments is the following. As we know, the pairwise distance in the Kullback–Leibler geometry and the Euclidean or Fisher one can be very different. Indeed, on the standard simplex, the former can approach infinity while it is bounded in the latter. Still, our experiments show that the geometric-topological structures—at least in low dimensions—do not typically show large discrepancies between the geometries. Is this also true in higher dimensions? How does this generalize to other Bregman divergences and the corresponding generalized Fisher distances [6]? To what extent can the Fisher space replace the Shannon space in various applications?

Several open questions about experimental understanding of geometric spaces arise:How do these results generalize to other notions of entropy, such as the Burg entropy [37]? The resulting Bregman Itakura–Saito divergence [38] is used to compare speech samples, but little is known about the resulting geometry and how it compares with other geometries.Is there a deeper reason for the similarity of the studied geometries? For example: are there Pinsker-type inequalities [39,40] between the pairwise dissimilarities (as well as other Bregman divergences)?Can the sensitivity of Delaunay mosaics to the dissimilarity be quantified probabilistically, as the expected Jaccard distance for random point processes?Persistence has been used before to compare metric spaces [41], and it would be interesting to know whether there are deeper connections to our work.Can the Fisher metric be extended beyond the positive part of the sphere? What is the geometry of the preimage of the Antonelli map in this case?

We finish with a technical question concerning the Delaunay mosaics in Fisher geometry: is the drawing we obtain by mapping the vertices to the corresponding points and connecting them with straight edges, flat triangles, etc., necessarily a geometric realization of the mosaic?

## Figures and Tables

**Figure 1 entropy-26-00637-f001:**
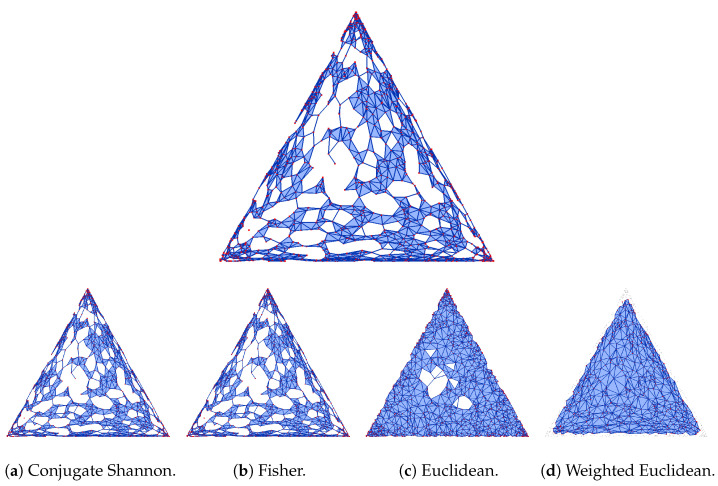
(**Top**) The Bregman–Alpha complex in Shannon geometry of a set *Y* of 1000 random points in $\Delta^{2}$ with threshold h=0.0025. (**Bottom row**) The reconstructions in four different geometries.

**Figure 2 entropy-26-00637-f002:**
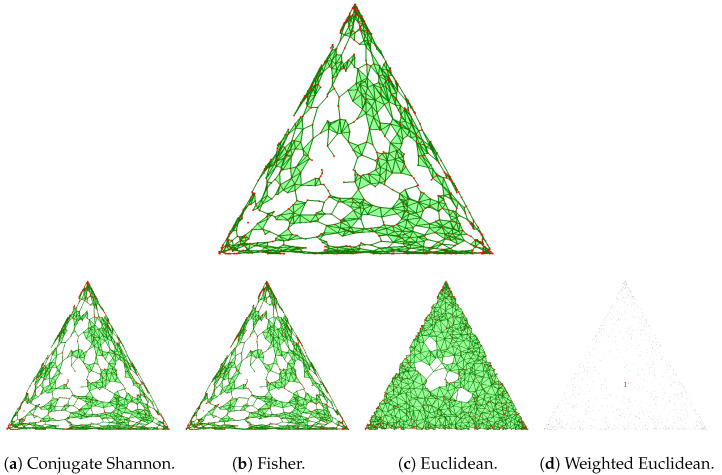
(**Top**) The Bregman–Wrap complex in Shannon geometry of the same points and threshold as in Figure 1. (**Bottom row**) The reconstructions in four different geometries.

**Figure 3 entropy-26-00637-f003:**
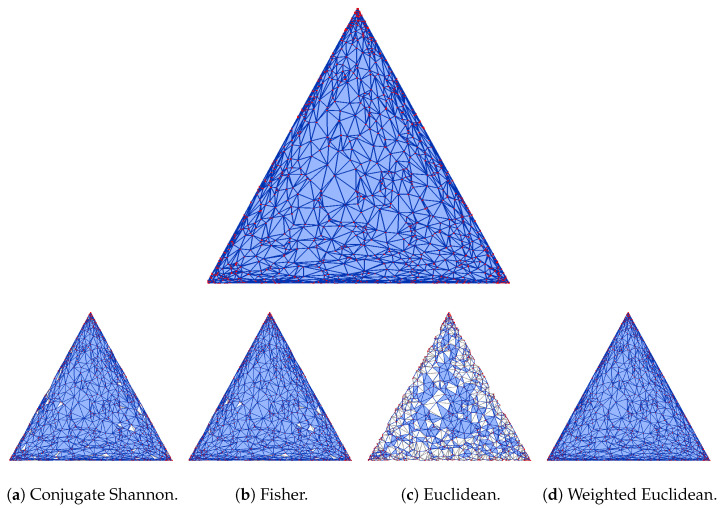
(**Top**) The Bregman–Delaunay mosaic in Shannon geometry for the same set of points as used in Figure 1 and Figure 2. (**Bottom row**) Four Delaunay mosaics whose triangles and edges are colored depending on whether or not they belong to the Shannon–Delaunay mosaic.

**Figure 4 entropy-26-00637-f004:**
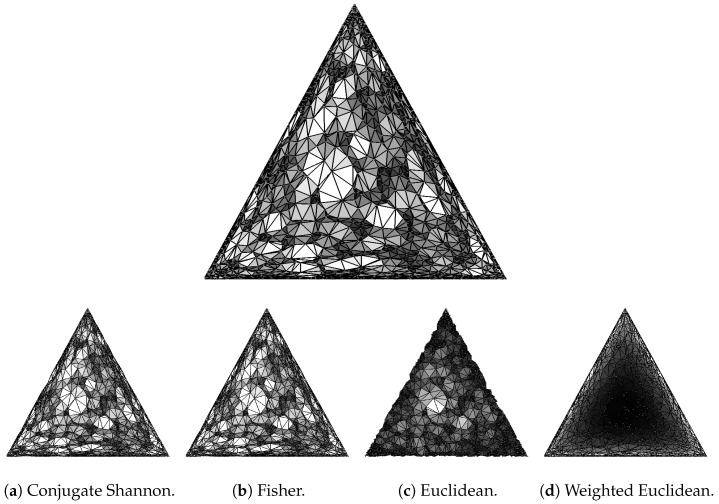
(**Top**) the color-coded Bregman–Delaunay mosaic in Shannon geometry of the same set of points as in Figure 3. (**Bottom row**) The color-coded Delaunay mosaics.

**Table 1 entropy-26-00637-t001:** The Jaccard distances between the Delaunay mosaics in Shannon, conjugate Shannon, Fisher, Euclidean, and weighted Euclidean geometries for points in the positive quadrant on the top and in the standard triangle on the bottom.

*J*	$D_{0}$	$D_{1}$	$D_{2}$	$D_{3}$	$D_{4}$
$D_{0}$	0.00	0.06	0.04	0.48	0.00
$D_{1}$		0.00	0.02	0.47	0.06
$D_{2}$			0.00	0.47	0.04
$D_{3}$				0.00	0.48
$D_{4}$					0.00
	$E_{0}$	$E_{1}$	$E_{2}$	$E_{3}$	$E_{4}$
$E_{0}$	0.00	0.10	0.06	0.52	0.00
$E_{1}$		0.00	0.04	0.51	0.10
$E_{2}$			0.00	0.51	0.06
$E_{3}$				0.00	0.52
$E_{4}$					0.00

**Table 2 entropy-26-00637-t002:** The bottleneck distances between the persistence diagrams of the rise functions on the Delaunay mosaics in Shannon, conjugate Shannon, Fisher, Euclidean, and weighted Euclidean geometries for points in the positive orthant on the top and points in the standard triangle on the bottom.

*B*	$d_{0}$	$d_{1}$	$d_{2}$	$d_{3}$	$d_{4}$
$d_{0}$	0.0000	0.0028	0.0004	0.0126	0.0048
$d_{1}$		0.0000	0.0028	0.0126	0.0048
$d_{2}$			0.0000	0.0126	0.0048
$d_{3}$				0.0000	0.0126
$d_{4}$					0.0000
	$e_{0}$	$e_{1}$	$e_{2}$	$e_{3}$	$e_{4}$
$e_{0}$	0.0000	0.0006	0.0003	0.0031	0.0035
$e_{1}$		0.0000	0.0003	0.0030	0.0034
$e_{2}$			0.0000	0.0030	0.0034
$e_{3}$				0.0000	0.0023
$e_{4}$					0.0000

## Data Availability

The software and data used for this study is available at: https://git.ista.ac.at/katharina.oelsboeck/wrap_2_3-public/ (accessed on 19 July 2024).

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
