# Peer review of "Understanding Higher-Order Interactions in Information Space"

_entropy, 2024, doi:10.3390/e26080637_

Round 1

Reviewer 1 Report

Comments and Suggestions for Authors

Summary. The authors explore higher-order interactions in information spaces using topological data analysis. They extend methodologies traditionally used in Euclidean spaces to information-theoretic settings like Bregman divergences. In so doing, they introduce new algorithms to analyze geometric-topological structures in these spaces, particularly comparing their performance versus Shannon and Fisher metrics. They implement these algorithms and demonstrate their application through experiments, highlighting differences in the Delaunay mosaics and Alpha shapes across some geometries.

Comments. This is a nice advance for the field of topological data analysis, extending traditional Euclidean methods to information-theoretic metrics like the Bregman geometry. They support their approach by applications to several geometric settings and provide valuable insights into the differences with existing methods. Their new algorithm is sound and the theoretical framework is well-developed. My main concern is that the presentation of the results is highly technical; as a result, the nice improvements made by the authors in their work will be likely appreciated only by experts in the topic. I would therefore recommend the authors to improve their presentation so to enhance the paper's clarity and its broad impact from the very beginning, perhaps devoting more breath to the applications and later indulging in a more detailed description of their methods. This is merely a suggestion aimed at broadening the possible readership interested in the authors’ work as well as in the field. Overall, I think the manuscript contains sufficient novelty and provides new methods for advancing research in topological data analysis.

Author Response

Thank you for the review and highlighting an important point regarding accessibility of this work.

Comment 1: I would therefore recommend the authors to improve their presentation so to enhance the paper's clarity and its broad impact from the very beginning, perhaps devoting more breath to the applications and later indulging in a more detailed description of their methods.

Response 1: While it would be hard to significantly improve the clarity of the technical portion of the paper, we extended the Introduction and Discussion sections. In particular, we are more explicit about the connections with applications, and provide a high level overview of the results. 

(Additionally, we improved the clarity of some of the technical parts thanks to detailed comments of one other reviewer.)

Reviewer 2 Report

Comments and Suggestions for Authors

In this work, the authors generalized two geometric-computational methods to Bregman and Fisher geometries, presenting an implementation of the methods and relevant computational experiments. I believe that the paper is well-written and the results sound, but there are a few issues that I hope the authors may address before their work can be published.

1) The references appear to be particularly selective. Only a few references to previous results are provided. I believe it would greatly help the reader if the authors could provide more background literature. A few examples: In line 28, standard references to Shannon and Fisher geometries are lacking; in line 32, no references are present to back the claim that machine learning models are trained with these geometries; in line 280, the Cech construction is mentioned for the first time in the paper without a reference; etc.

2) In line 126 the authors introduce the negative Shannon entropy as E(x) = \sum_{i = 1}^d [x_i \log x_i - x_i]. Unless I'm missing something, this definition (and the corresponding definition of relative entropy in Eq. (9)) is a bit odd. Let's say that x_i is a point in the simplex (line 135 of the paper).  Then, the Shannon entropy would be defined as - \sum_{i = 1}^d x_i \log x_i. Where does the extra term come from? The lack of references here is particularly relevant, as it is not clear why the authors do not use the standard definition.

3) I believe that it would help the reader and subsequent investigations if the authors could share the code used to perform the computational experiments.

4) It would be helpful if, in the Discussion, the authors could provide some concrete future applications of the different geometries in other contexts (they mention machine learning algorithms in the introduction), highlighting their benefits or drawbacks.

5) Some minor issues:

a) The appendixes start from H in my manuscript. This is an odd choice, does the manuscript miss some appendixes?

b) In eq. (1), I assume that <> denotes the inner product, but the authors may want to define the notation for completeness.

c) Line 91: "The" misses the capitalization at the beginning of the sentence.

d) Line 130: I could not find in reference (5) the result for the conjugate of the negative Shannon entropy.

e) Line 420: "He" should be "We" I believe.

Author Response

Thank you for the detailed comments. We fixed all the issues, which improved the readability of the paper.

Comment 1: The references appear to be particularly selective. Only a few references to previous results are provided. I believe it would greatly help the reader if the authors could provide more background literature [...]

Response 1: We completely agree. We added 18 references, and made sure that key concepts now come with a citation. 

Comment 2: In line 126 the authors introduce the negative Shannon entropy as E(x) = \sum_{i = 1}^d [x_i \log x_i - x_i]. Unless I'm missing something, this definition (and the corresponding definition of relative entropy in Eq. (9)) is a bit odd. Let's say that x_i is a point in the simplex (line 135 of the paper).  Then, the Shannon entropy would be defined as - \sum_{i = 1}^d x_i \log x_i. Where does the extra term come from? The lack of references here is particularly relevant, as it is not clear why the authors do not use the standard definition.

Response 2: That's a good point, which we now clarify in the paper. In short: this definition does not affect the resulting divergence, but simplifies some further computations. (In particular the gradient is slightly simpler.) (Prompted by the comment we also clarified the formula for the KL divergence, which is again slightly different than the standard one. They agree on the standard simplex, but ours allows us to work also on the entire positive orthant of R^d, which is useful for some of the synthetic experiments.)

Comment 3: I believe that it would help the reader and subsequent investigations if the authors could share the code used to perform the computational experiments.

Response 3: Yes, the code is now available and we provide an url in the paper (page 2, footnote).

Comment 4: It would be helpful if, in the Discussion, the authors could provide some concrete future applications of the different geometries in other contexts (they mention machine learning algorithms in the introduction), highlighting their benefits or drawbacks.

Response 4: Agreed. We expanded the discussion related to data science/machine learning. We onw mention one other information-theoretical geometry (the Itakura-Saito divergence induced by the Burg entropy) used in speech analysis. 

Comment 5) Some minor issues:

All fixed, thank you for the careful read through!

Comment 5d: Line 130: I could not find in reference (5) the result for the conjugate of the negative Shannon entropy.

Response to 5d: We rewrote the sentence clarifying that we mean that equation (5) is used to derive the conjugate.